# A Linear Programming Model for Operational Optimization of Agricultural Activity Considering a Hydroclimatic Forecast—Case Studies for Western Bahia, Brazil

Igor Boninsenha [1,*], Everardo Chartuni Mantovani [1], Marcos Heil Costa [1] and Aziz Galvão da Silva Júnior [2]

1   Department of Agricultural Engineering, The Federal University of Viçosa, Viçosa 36570-900, MG, Brazil
2   Department of Rural Economy, The Federal University of Viçosa, Viçosa 36570-900, MG, Brazil
*   Correspondence: igor.boninsenha@ufv.br; Tel.: +55-31-99908-0875

**Abstract:** The water crisis is a reality in Western Bahia. In this region, a hydroclimatic monitoring system capable of providing water availability information in advance for water users was implemented objectively to avoid water conflicts. In this study, we proposed the integration between the hydroclimatic monitoring system and a linear programming method to optimize the agricultural net benefit considering the scenarios of rainfall delay or reduction. Case studies were demonstrated in five farms and three municipalities of Western Bahia. The results show that in irrigated areas, the model optimizes the net economic benefit by the possibility of a continuous or double-cropping system, even in rainfall delay or reduction, where irrigation can supply the water demand of crops. In rainfed areas, it is noticeable that in rainfall delay or reduction scenarios, the model response is not to crop, due to the accentuated water deficit in crops, which may cause a significant yield reduction. It was found in a farm-level analysis, where the model response was not to crop, and farmers had a yield reduction of 61% in rainfed areas. This study opens the integration between the optimization methodologies and the hydroclimatic monitoring system with new insights into how this integration can guide water governance actions in regions where the water crisis is a reality.

**Keywords:** water governance; climate change; water security



## 1. Introduction

One of the main risks for society in the following years is the water crisis [1]. The projections of population increase [2], climate change [3], and the attribution to agriculture are mainly responsible for global water withdrawal [4], supporting this concern. Thus, global food production must increase in uncertain scenarios of water resources, and without local actions, the water crisis scenarios may become a reality rather than a risk.

Recent estimates by the Pacific Institute [5] show that global water conflicts have increased. These conflicts are reported mainly in developing countries [6–9] with a solid agricultural dependency on national economies. Gilbert [10] declared that to avert a global water crisis, we need to have better data related to water demand and availability and use these data for assertive management decisions.

The relationship between agriculture and the water crisis in Brazil is evident in a specific mesoregion, Western Bahia. Western Bahia is located in MATOPIBA (the acronym stands for the first two letters of the Brazilian states of Maranhão, Piauí, Tocantins, and Bahia), which is considered the last agricultural frontier in the country. The region is characterized by rapid changes in land cover and land use, predominantly over the natural vegetation of the Cerrado biome, through significant agricultural intensification [11]. The agrarian model practiced in Western Bahia has an impressive performance in the regional economy. The 2.4 million hectares destined for crop production concentrate 34.2% of the state agricultural gross domestic product [12]. This agricultural model is well known for the high concentration of irrigated areas. Despite covering only 5.81% of the cropland, they

are responsible for approximately 30% of its agricultural production value [13]. However, despite the economic benefits of irrigated areas, these are the leading cause of water stress and conflicts in Western Bahia [14]. On the other hand, most of the croplands in the region are rainfed, with the adoption of intensive double-cropping systems in some areas where the rainy season is long enough. However, the sizeable interannual variability on the onset of the rainy season and an observed shortening of its duration significantly increases the risk of such intensive practices [15].

To avoid water conflicts in Western Bahia, Pousa et al. [14] and Mantovani et al. [16] suggested a hydroclimatic monitoring system capable of predicting water availability in the months of water scarcity and providing information about the surface and groundwater availability, land use, hydrological features, policy, and governance. A prototype of this system is available on the OBahia—Territorial and Water Intelligence Platform for Western Bahia (http://obahia.dea.ufv.br, accessed on 7 July 2020) [17]. For the information available on the OBahia platform to become a guide for assertive management decisions, we need to integrate this system with data analysis tools. For example, the "rainy season onset forecast" tool provides information that can be useful for planning the crop season, helping farmers to decide the best sowing dates for each crop, for resource allocation during crop season, and predicting harvest losses. However, without an effective tool for analyzing these data, this type of forecast would only be another piece of information whose usefulness would not be used by the system's users.

Several studies have been developed to optimize agricultural output. Linear programming (LP) has superb usability in these studies, with the problem formulation adapted to different objectives such as (1) maximizing net economic benefits [18]; (2) reducing crop production costs [19,20]; (3) indicating resources allocation [21–27]; and (4) improving economic and environmental indicators [28,29]. The LP method was also demonstrated in different management scales, with multiple objectives, under parameter uncertainty, small data series, and various other conditions [30–34]. Despite the significant contribution of LP models, the integration of models with hydroclimatic forecast models is non-existent. This combination can favor agricultural water management in a region, as the forecast results can be analyzed automatically, allowing for in-depth analysis and decision making at different management levels.

Considering the hydroclimatic forecast provided for Western Bahia, it becomes necessary to study how the LP methods can be applied to improve the water management in the region, guiding farmers to better crop decisions and regional water authorities to effective management strategies. This study proposes an LP model for operational optimization in Western Bahia that is fully integrable with the hydroclimatic monitoring system information. The model optimizes agricultural output considering the in-farm restrictions and water availability that depend on the rainy season onset forecast. Specifically, this work formulates an LP model and solves it for applications at distinct water availability and use levels. It also indicates strategies to make the model application effective. The work structure is divided to describe the region, present how the dataset was acquired and processed, how the model was formulated, present the case studies, and discuss how the application can be helpful for farmers and regional water authorities.

## 2. Materials and Methods

This section describes the acquisition and processing of the dataset and the linear programming model (LPM) formulation. In Figure 1, the complete work flowchart also shows the steps adopted in the case studies, results, and discussion sections.

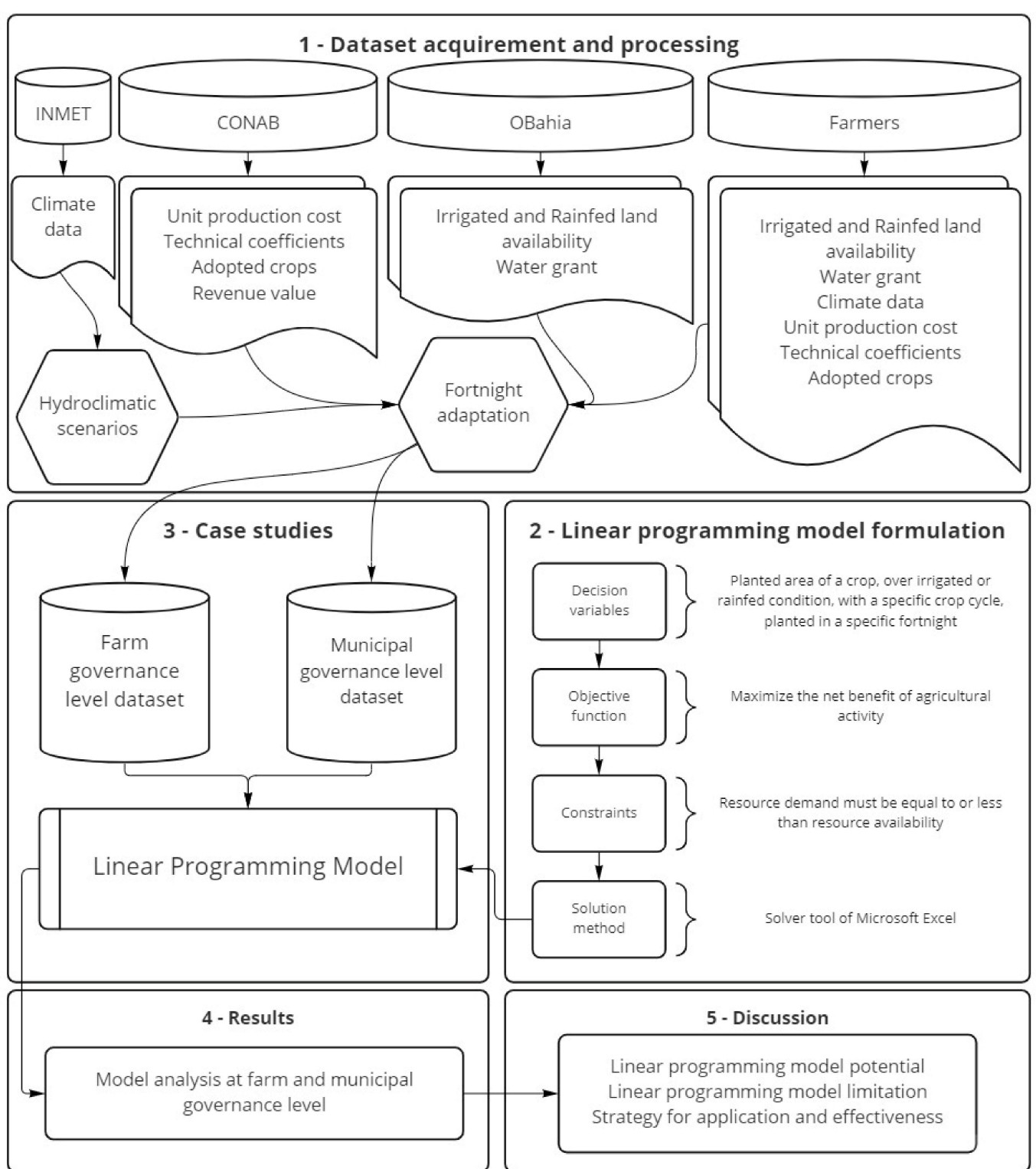

**Figure 1.** Work flowchart.

*2.1. Region Description*

Western Bahia (43.23° to 46.61° W, 10.10° to 15.26° S), represented in Figure 2, borders the Brazilian states of Minas Gerais, Goiás, Tocantins e Piauí. The region's representative climate is tropical with a dry season, which favors the development of irrigated agriculture using the center pivot system combined with the flat topography. From 2000 to 2020, the planted area of soybean, maize, cotton, and bean represented 94% ± 2% of the total

cultivated area in Western Bahia [35]. According to Pimenta et al. [36], in 2020, the total agricultural area was equal to $5.14 \times 10^6$ ha, and irrigated cropland was equal to $2.18 \times 10^5$ ha. The water grant in this region follows Federal Law n° 9.433/97 and State Law n° 11.612/09, conceding the use right of water resources by limited time to the applicant. This process in Western Bahia is coordinated by INEMA (Environment and Water Resources Institute of Bahia), which considers, for superficial withdrawal of 80% of $Q_{90}$, the minimum amount of water flow in a river that is present 90% of the time as the maximum of water grant for each river basin.

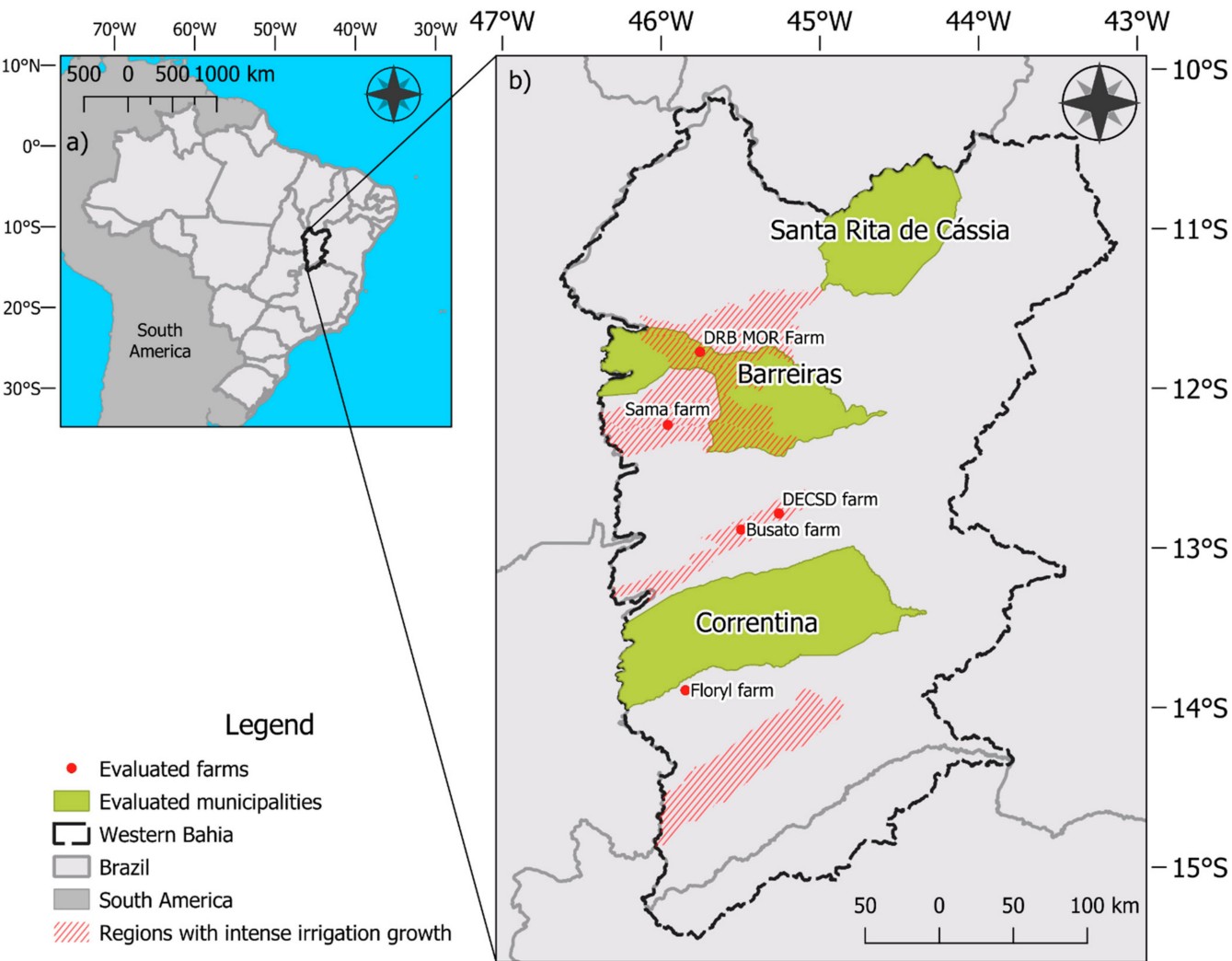

**Figure 2.** General view of Western Bahia region. (**a**) This region is defined by the union of basins of the Grande, Corrente, and Carinhanha Rivers. Part of the Carinhanha River basin is over the territory of Minas Gerais State. (**b**) Location of farms and municipalities where the linear programming model was applied in case studies (characterized in Section 3.1). Red dots represent the farms' locations. Green areas correspond to municipal areas. Red lines pattern fill represent areas more susceptible to the water crisis in Western Bahia, evaluated by Pousa et al. [14].

### 2.2. Dataset Acquisition and Processing

The data acquisition process depends on the level of water governance considered in the case studies section. At the farm governance level, through interviews and local consultation, obtained data are related to irrigated and rainfed area availability, water grant, weather data, unit production cost, technical coefficients (unit need for machinery and manual labor), and adopted crops. At the municipal governance level, weather data were obtained from "*Banco de Dados Meteorológicos do INMET*" [37], selecting at least 30 years of

daily data for each weather station. Unit production cost, technical coefficients, adopted crops, and revenue value were obtained from "*Série Histórica de Dados*" [38]. Irrigated and rainfed area availability and water grant were obtained from the OBahia platform [17]. Each crop's unit revenue value was obtained from "*Preços agrícolas, da sociobio e da pesca*" [39]. Weather data refer to both cases' maximum and minimum temperature, insolation, average relative humidity, and average wind speed.

At the farm level, the LPM was applied considering the original weather condition and in four scenarios of rainfall delay (D1, D2, D3, and D4, representing 1, 2, 3, and 4 fortnights in the onset of the rainy season, respectively) and rainfall reduction (R5, R10, R15, R20 representing 5%, 10%, 15%, and 20% of rainfall reduction, respectively). These scenarios were adopted to represent the possible responses of the hydroclimatic forecast. At the municipal level, the weather data were analyzed to determine the average and quartiles (Q25 and Q75 representing the 25% and 75% quantiles, respectively) for each fortnight to represent each municipality's rainfall condition; average values of weather data represented the original climatic condition. In addition to analyzing rainfall delays and reductions, the LPM was also applied for quartile conditions. When available, the actual area occupation was compared with the results of the LPM application. In both cases, the resource availability and needs were converted to represent the fortnightly period, making the model application compatible with the hydroclimatic forecast available for the region on the OBahia platform [17]. All characteristics of the LPM application are fully described in the case studies section.

*2.3. LPM Formulation*

The LPM formulation is fully described by Vanderbei [40] and Luenberger and Ye [41]. An LPM model is applied to optimize a process considering a linear correspondence of variables. In other words, it is a method to optimize linear processes subject to actual decision variables and constraints. This subsection highlights the definition of decision variables, the objective function, constraints, and the solution method.

2.3.1. Decision Variables

This study's decision variables (DV) are defined as the planted area of a crop, over irrigated or rainfed condition, with a specific crop cycle, produced in a particular fortnight. Mathematically, DV can be demonstrated, as $X_{ijkl}.X$ (in hectares) is the optimum value for the planted area; $i$ is the crop type, which may be soybeans, maize, beans, wheat, or cotton; $j$ is the planting system, which may be irrigated or rainfed; $k$ is the crop cycle type, which may be short, average, or long; and $l$ is the sowing period, limited by agroclimatic zoning made by the Brazilian National Supply Company (CONAB) [42]. The crop cycle type refers to the period between the sowing and harvest of the crop, defined by the genetics of the adopted variety.

2.3.2. Objective Function

The objective function (OF) of this optimization model is to maximize the net benefit (NB) (BRL ha$^{-1}$) of agricultural activity, which has been mathematically expressed as:

$$\text{Maximize NB} = \sum R_{ijkl} \times X_{ijkl} - \sum C_{ijkl} \times X_{ijkl}, \qquad (1)$$

where $R$ (BRL ha$^{-1}$) is the unitary revenue value, and $C$ (BRL ha$^{-1}$) is the unitary production cost.

2.3.3. Constraints and Solution Method

The constraints are based on the agricultural use of resources (Res) divided into land, work, machines, and capital. Land constraints refer to area (irrigated or rainfed) occupation; work constraints refer to labor and administrative activities; machines constraints refer to agricultural machinery's availability as a planter or harvester; capital constraints mean the working capital of activity. In all cases, the resource demand was computed but was

used as constraints when that information was substantial for the management level; all constraints' setups are defined in the case study section. In all constraint conditions, the use of the resources must be equal to or less than availability. In general, the resource demand is described by Equation (2) as the product between the optimum DV and the unitary resource amount needed.

$$\text{Res demand} = \sum X_{ijkl} \times \overline{Y}_{ijkl}, \tag{2}$$

where $\overline{Y}$ (dimensionless) is the adopted technical coefficient for each resource, representing the unitary amount needed.

Crop water demand (CWD) was the only constraint parameter calculated differently than described by Equation (2). CWD is calculated in steps, first, the reference evapotranspiration ($ETo$, mm day$^{-1}$) and the crop evapotranspiration ($ETc$, mm day$^{-1}$) accumulated for the period. $ETc$ is the product between $ETo$ and the crop coefficient ($Kc$, dimensionless), as indicated by Allen et al. [43] and described in Equation (3).

$$ETc= ETo \times Kc_{ijkl} \tag{3}$$

In Cerrado's condition (Cerrado's region includes Western Bahia), the reductions of 20% in $ETc$ for cotton and soybean, 12% for maize, and 13% for beans do not cause a significant reduction in crop yields [44]. This work defined CWD as the product between the related $ETc$ and a potential crop evapotranspiration reduction that does not cause a significant yield reduction ($Kr$, adimensional). In turn, the adopted $Kr$ for each crop was 0.80 for cotton and soybean, 0.88 for maize, 0.87 for beans, and 0.85 for wheat. Therefore, CWD for each DV is mathematically described by Equation (4).

$$CWD_{Xijkl}= \sum X_{ijkl} \times ETc \times Kr \tag{4}$$

The Solver tool of Microsoft Excel was used to set all the LPM applications. The "LP Simplex" was the solution method applied on all occasions.

## 3. Case Studies

### 3.1. Farms and Municipalities Characterization

Due to data availability, five farms served as case studies at the farm governance level: (1) Sama, where the LPM was applied to optimize the Soybean crop season with different cycle durations and sowing dates in rainfed and irrigated areas; (2) Floryl, for multiple crops with multiple cycle duration, (3) DRB MOR ("Decisão Rio Branco e Morena Farm") and (4) DECSD ("Decisão São Desidério Farm") for multiple crops during six seasons and finally (5) Busato, for multiple crops in a season. Sama, DRB MOR, DECSD, and Busato farms are included in the regions of intense irrigation systems growth presented by Pousa et al. [14]. Three municipalities served as case studies at the municipal governance level: (1) Barreiras, with the largest irrigated area; (2) Correntina, a region with the largest rainfed agricultural area; and (3) Santa Rita de Cássia, a region with the lowest technology agriculture. In all cases, cotton, beans, soybeans, and maize were evaluated during six crop seasons. For the study sites, the resources' availability, crops, and season are summarized in Table 1.

**Table 1.** Description of resource availability, crops, and crop season adopted in LPM's analysis in municipalities and farms.

| Farm or Municipality | Irrigated Area | Rainfed Area | Water Grant | Evaluated Crop | Season | Labor Availability | Tractor Availability | Truck Availability | Sprayer Availability | Harvester Availability |
|---|---|---|---|---|---|---|---|---|---|---|
| | (Hectares) | (Hectares) | (m$^3$ day$^{-1}$) | | | (h day$^{-1}$) | (h day$^{-1}$) | (h day$^{-1}$) | (h day$^{-1}$) | (h day$^{-1}$) |
| Sama [f] | 1221.10 | 878.50 | 39,297.50 | Soybean | 2018/19 | 160 | 16 | 16 | 32 | 16 |
| Floryl [f] | 950.00 | 120.00 | 68,437.00 | Maize [1−2]; Soybean | 2019/20 | | | | | |
| DRB MOR [f] | 4267.00 | 646.00 | 101,760.00 | Cotton; Bean [1−2]; Maize [1−2]; Soybean; Wheat | 2017 [aw]; 2017 [ss]; 2018 [aw]; 2018 [ss]; 2019 [aw]; 2019 [ss] | | | | | |
| DECDS [f] | 3299.00 | 0.00 | 156,552.00 | | | | | | | |
| Busato [f] | 4246.80 | 676.00 | 23,240.00 | Cotton; Maize [1−2]; Soybean | 2019/20 | | | Not applicable | | |
| Barreiras [m] | 42,760.00 | 291,948.00 | 4,586,256.52 | | | | | | | |
| Correntina [m] | 15,008.00 | 408,022.00 | 2,444,307.23 | Cotton; Bean [1−2]; Maize [1−2]; Soybean | 2018 [ss]; 2019 [aw]; 2019 [ss]; 2020 [aw]; 2020 [ss]; 2021 [aw] | | | | | |
| Santa Rita de Cássia [m] | 60.00 | 7293.00 | 44,000.00 | | | | | | | |

[f] Represents the farms, [m] represents the municipality, [aw] means the autumn/winter season, [ss] illustrates the spring/summer season, and [1−2] represents the crop cultivated in 1st or 2nd season in a double-cropping system. At Sama farm, the labor, tractor, truck, sprayer, and harvester availability, in h day$^{-1}$ represents the total resource availability in one day, given the product between the amount of machinery or employees and the work journey in a day, equal to eight hours.

The minimum and maximum values of *ETo* and rainfall observed for each farm are presented in Table 2.

**Table 2.** Minimum and maximum rainfall values observed for each farm or municipality.

| Farm or Municipality | Minimum *ETo* | Maximum *ETo* | Minimum Rainfall | Maximum Rainfall |
|---|---|---|---|---|
| | (mm fortnight$^{-1}$) | (mm fortnight$^{-1}$) | (mm fortnight$^{-1}$) | (mm fortnight$^{-1}$) |
| Sama [f] | 45.2 | 96.9 | 0.0 | 172.8 |
| Floryl [f] | 42.7 | 86.7 | 0.0 | 142.8 |
| DRB MOR [f] | 46.2 | 107.5 | 0.0 | 142.8 |
| DECDS [f] | 43.1 | 89.9 | 0.0 | 232.2 |
| Busato [f] | 41.1 | 94.7 | 0.0 | 168.8 |
| Barreiras [m] | 54.7 | 88.6 | 0.2 | 103.7 |
| Correntina [m] | 54.5 | 84.8 | 0.2 | 108.7 |
| Santa Rita de Cássia [m] | 59.0 | 91.6 | 0.0 | 101.5 |

[f] Represents the farms, [m] represents the municipality.

### 3.2. Crop Characterization

Table 3 shows the adopted characteristics for each crop. These data are usually adopted by companies that provide irrigation consulting services, such as Valley Scheduling® (called in Brazil Irriger Connect®). The adopted technical coefficients, as labor and machinery needs for each condition, and the adopted revenue values are demonstrated in Supplementary Materials S1.

### 3.3. Model's Constraints Setup

For each application and LPM analysis, a different constraint setup was adopted. These constraints may be the irrigated (Ia) or rainfed area (Ra), CWD irrigated (CwdI) or rainfed areas (DwdR), labor (L), or machinery (M). This setup difference was necessary due to the solution tool's limitations and can be thoroughly checked in Table 4.

**Table 3.** Crop, cycle type, duration *Kc*, *Kr*, and ranging sowing time.

| Crop | Cycle Type | Cycle Duration (Fortnight, or 15 Days) | Initial *Kc* | Average *Kc* | Final *Kc* | *Kr* [44] | Range Sowing Time (Fortnight–Month) [42] |
|---|---|---|---|---|---|---|---|
| Soybean | short<br>average<br>long | 8<br>9<br>10 | 0.60 | 0.70 | 0.80 | 0.80 | 1–10 to 2–02 |
| Maize 1st season | average<br>long | 9<br>12 | 0.65<br>0.60 | 1.00 | 0.60<br>0.50 | 0.88 | 1–10 to 2–02 |
| Maize 2nd Season | average<br>long | 9<br>12 | 0.65<br>0.60 | 1.00 | 0.60<br>0.50 | 0.88 | 1–05 to 2–06 |
| Cotton | long | 14 | 0.50 | 0.90 | 0.38 | 0.80 | 1–11 to 2–02 |
| Bean 1st season | average | 7 | 0.70 | 1.20 | 0.60 | 0.87 | 1–10 to 2–02 |
| Bean 2nd season | average | 7 | 0.70 | 1.20 | 0.60 | 0.87 | 1–04 to 2–06 |
| Wheat | average | 8 | 0.70 | 1.20 | 0.40 | 0.85 | 1–08 to 2–09 |

**Table 4.** Constraint setup for each LPM application and analysis.

| Analysis | Farms | | | | | Municipalities | | |
|---|---|---|---|---|---|---|---|---|
| | Sama | Floryl | DRB MOR | DECSD | Busato | Barreiras | Correntina | Santa Rita de Cássia |
| Original | Ia; Ra; CwdI | - | - | - | - | Ia; Ra; CwdI; CwdR | Ia; Ra; CwdI; CwdR | Ia; Ra; CwdI; CwdR |
| Irrigated original Irrigated rain delay Irrigated rain reduction | Ia; CwdI; L; M | Ia; CwdI | Ia; CwdI | Ia; CwdI | Ia; CwdI | Ia; CwdI | Ia; CwdI | Ia; CwdI |
| Rainfed original Rainfed rain delay Rainfed rain reduction | Ra; CwdR; L; M | Ra; CwdR | Ra; CwdR | - | Ra; CwdR | Ra; CwdR | Ra; CwdR | Ra; CwdR |

Ia means irrigated area demand, Ra means rainfed area demand, CwdI means crop water demand in irrigated areas, CwdR means crop water demand in rainfed areas, L means labor demand, and M means machinery demand.

## 4. Results

This section presents the results of applying the LPM for the Western Bahia farms and municipalities.

### 4.1. LPM Application at Farms

At Sama farm, LPM results differ from real decisions made by farmers. In irrigated areas, the LPM's sowing recommendation was later compared to the actual conditions, while the harvest indication was similar to the real condition. In rainfed areas, without considering crop water demand as a constraint, only a tiny difference in the harvest date is noticeable in comparing the LPM and actual condition due to changes in the crop cycle. These results are demonstrated in Figure 3.

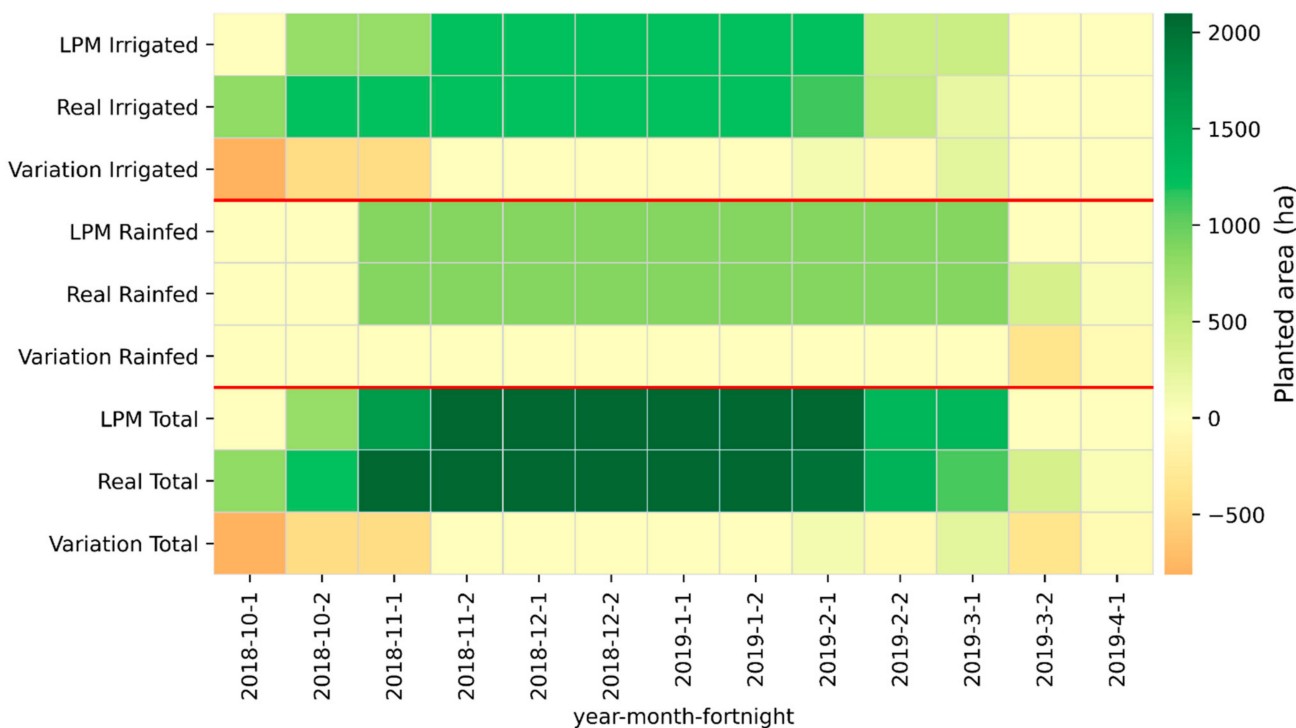

**Figure 3.** Planted area, in hectares, recommended by LPM application for irrigated and rainfed areas of Sama farm.

In irrigated areas, considering rainfall delay conditions, the LPM's responses considered the best price for crop commercialization and distribution of labor and machinery. The LPM's response considering rainfall reduction scenarios did not differ between them because irrigation can supply the crop water demand during the crop season. When considering the crop water demand in rainfed areas, the LPM's response was not to plant. This recommendation occurred because the crop's water stress, in simulated conditions, was higher than the one proposed by Santos et al. [44], which could cause yield losses. According to Sama farm managers, the crops in this harvest suffered an average yield reduction of 12% and up to 61% (considering the expected and actual) in rainfed areas. The LPM's application results for irrigated areas are demonstrated in Figure 4. Results for rainfed areas were omitted due to the null LPM response.

At Floryl farm, the LPM optimizes (in all conditions of rainfall delay or reduction) the area occupied in irrigated areas recommending the sowing of soybeans (from October to February) followed by maize first crop (from February to June) and then by maize second crop (from June to October). Double-cropping systems are standard in Brazil, where almost 58% of maize is produced under these conditions [15], but the adoption of continuous monoculture (maize first and second crop) may not be recommended due to the increase in pathogen population in the area, which can cause yield reduction of the second crop [45]. In rainfed areas, the LPM's response was to plant maize first crop on all occasions and adjust the sowing date in rainfall delay conditions. Sowing at the D4 scenario was not recommended due to the CONAB's sowing calendar [42]. This result is also consistent with the assessment made by Abrahão and Costa [15], where rainfall delay is one of the threats to double-cropping systems. All LPM results of area occupation can be verified in Figure 5.

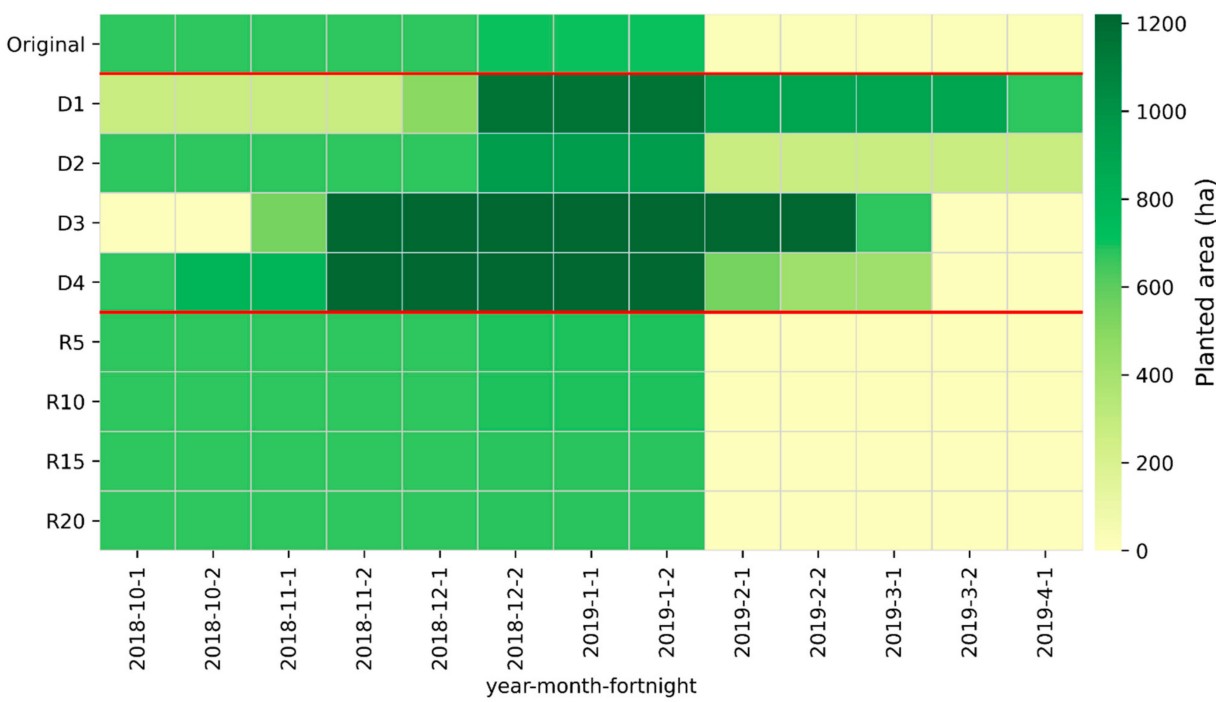

**Figure 4.** Planted area, in hectares, recommended by LPM application for irrigated areas of Sama farm considering rainfall delay and reduction scenarios.

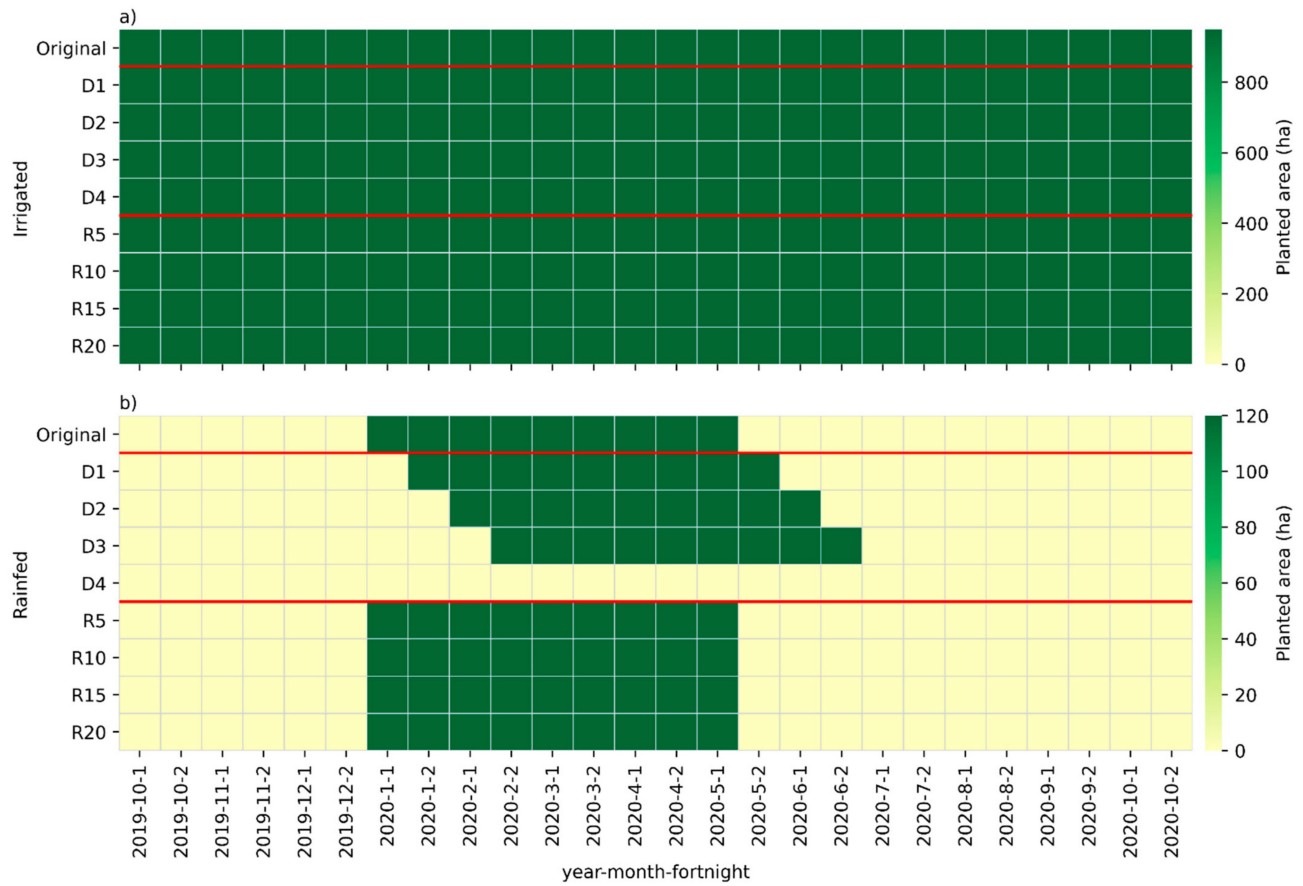

**Figure 5.** Planted area, in hectares, recommended by LPM application at Floryl farm. (**a**) Shows the results for irrigated areas. (**b**) Shows the results for rainfed areas.

The results for the DRB MOR farm were similar to the Floryl farm. In irrigated areas, the LPM's responses favor higher area occupation by sowing cotton and bean second crops during November 2017 and September 2018, followed by sowing bean first crop, cotton, and wheat during November 2018 and January 2020, in all scenarios of rainfall delay or reduction. In rainfed areas, the recommendation was cotton after September 2019 and December 2020, adjusting the sowing dates according to rain delay conditions. It is perceptible that from 2017 to 2019, there were no recommended crops in rainfed areas due to the accentuated water deficit. This response shows that crop yield in rainfed areas can be significantly reduced, especially in a drier year. At the DECSD farm, rainfed areas are not cultivated, and the LPM responses are similar to those obtained for the DRB MOR farm. LPM results for the DRB MOR and DECS farms are shown in Figure 6.

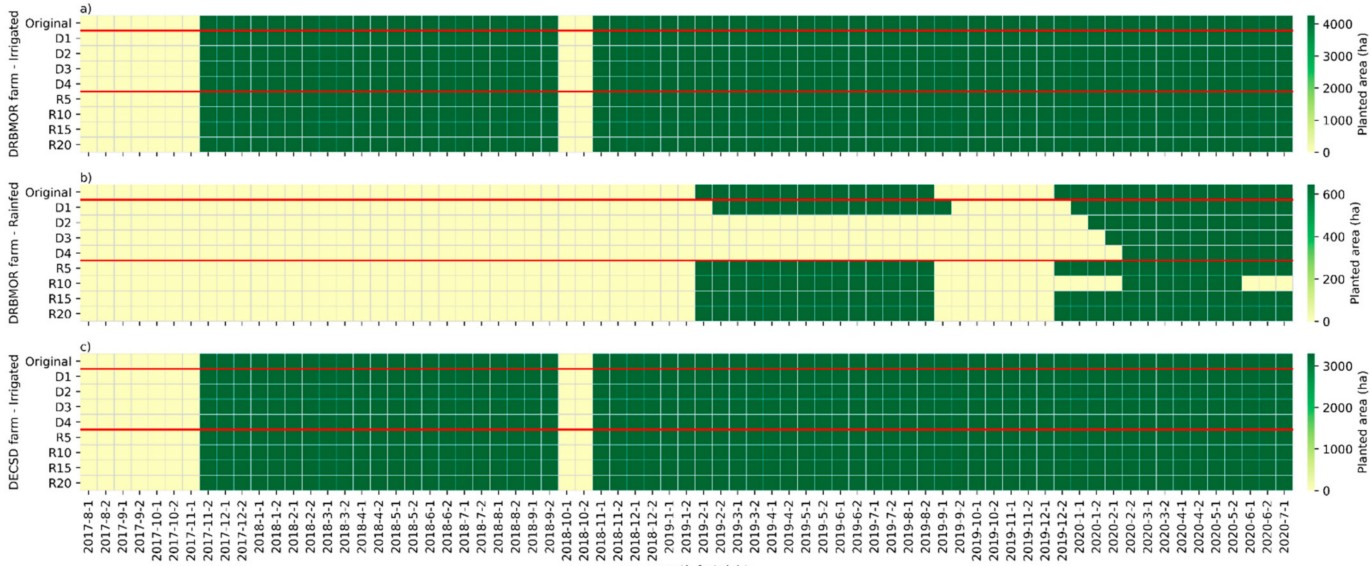

**Figure 6.** Planted area, in hectares, recommended by LPM application at DRB MOR and DECSD farms. (**a**) Shows the results for irrigated areas of DRBMOR farm. (**b**) Shows the results for rainfed areas of the DRB MOR farm. (**c**) Shows the results for irrigated areas of DECSD farm.

At the Busato farm, in irrigated areas, the LPM recommends a double-cropping system composed of cotton and maize second crops for all occasions of rainfall delay or reduction. In rainfed areas, only cotton is recommended, adjusting the sowing date in rainfall delay scenarios. Again, the double-cropping system is threatened by accentuated water deficit, as pointed out by Abrahão and Costa [15]. All results are shown in Figure 7.

### 4.2. LPM Application at Municipalities

Results of the LPM application at Barreiras show an excellent optimization of area occupation. The predominant recommendation is the double-cropping system, with beans first crop and cotton over irrigated and rainfed areas. Irrigated areas maintain the same results for all evaluated conditions. Over rainfall delay conditions, rainfed lands cannot be capable of a double-cropping system. In extremely dry year scenarios (Q25), the LPM recommendation is not to plant in rainfed areas, indicating a possibility of a decline in crop yield. Correntina and Santa Rita de Cássia results resembled the Barreiras results, with cotton and beans in a double-cropping system for irrigated areas. The model indicated an accentuated water deficit in rainfed areas that may cause a significant yield reduction, threatening the double-cropping systems.

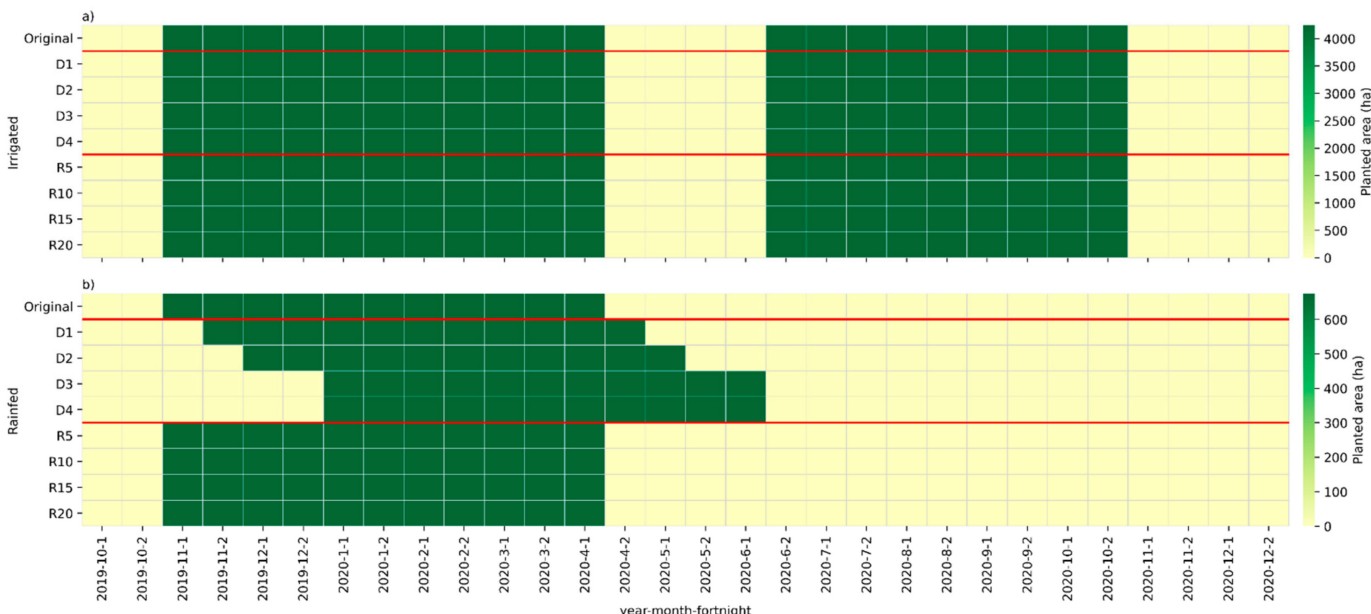

**Figure 7.** Planted area, in hectares, recommended by LPM application at Busato farm. (**a**) Shows the results for irrigated areas. (**b**) Shows the results for rainfed areas.

Among those evaluated, Barreiras is the municipality with the most irrigated area to total cropland (12.78%). On the other hand, Correntina and Santa Rita de Cássia have smaller proportions (3.54% and 0.81%, respectively). Despite not being in regions of intense irrigation growth [14], Correntina has been the scene of conflicts over water resources [46]. Santa Rita de Cássia and other regions not evaluated in the study have a vast potential for irrigated agriculture development. Still, suitability and technical limitations must be considered (as energy availability, topography, climatic conditions, and logistical aspects), and beyond these, as well as the development's environmental, economic, and social implications. The complete results of the LPM application to Barreiras, Correntina, and Santa Rita de Cássia are shown in Figure 8.

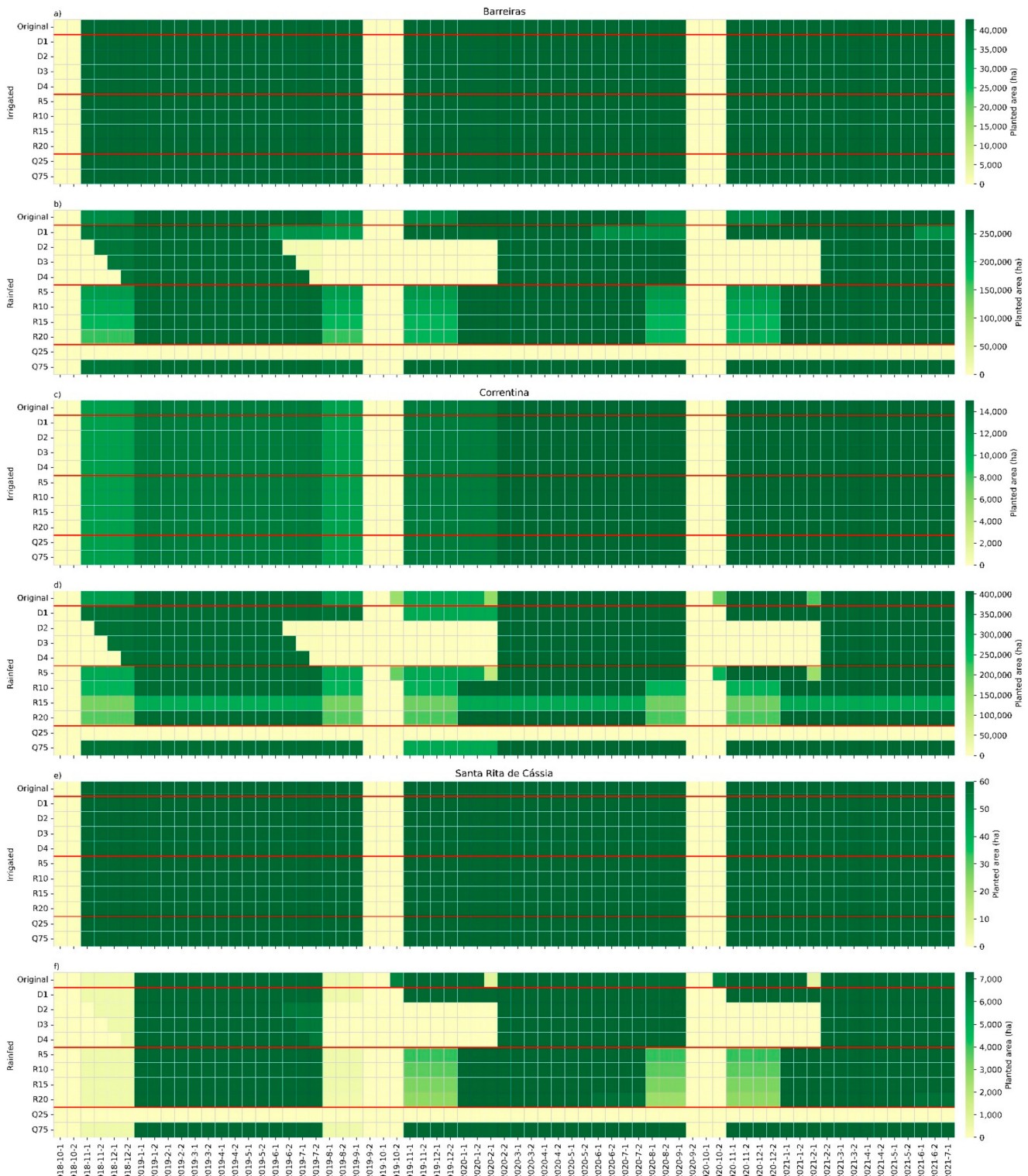

**Figure 8.** Planted area, in hectares, recommended by LPM application for irrigated and rainfed conditions of Barreiras, Correntina, and Santa Rita de Cássia. (**a**) Shows the results for irrigated areas of Barreiras. (**b**) Shows the results for rainfed areas of Barreiras. (**c**) Shows the results for irrigated areas of Correntina. (**d**) Shows the results for rainfed areas of Correntina. (**e**) Shows the results for irrigated areas of Santa Rita de Cássia. (**f**) Shows the results for rainfed areas of Santa Rita de Cássia.

## 5. Discussion

This work presents an integrated approach between the hydroclimatic monitoring system and a linear programming model to guide actions of water governance in Western Bahia. Given the proposed model formulation, optimizing agricultural activity's net economic benefit in rainfall delay and reduction scenarios was possible. This optimization was made by the recommendation of crops, cycle type, and sowing dates, considering the limitations of water, land, capital, and labor of each farm or municipality. In this way, it becomes possible to plan a crop season based on information on water availability in the region and avoid significant water stress in crops. This information is substantial to proactive water management decisions in Western Bahia.

The presented results in this work are consistent with those found in other studies. There was success in applying the model at different levels of water use, such as farms and municipalities, in the same way, demonstrated by several authors [21–27]. At the farm level, it becomes perceptible that the model can recommend or not crops, given the water availability scenarios. In the irrigated area, the model response is generally directed to select the crop with the maximum net benefit. In rainfed areas, the model recommendation was not to crop occasionally. It occurred due to the expected water deficit in a crop season or to the scenarios of rainfall delay when the beginning of the rainy season is out of the sowing calendar. This occurrence was noticed in the Sama farm simulation, where considering the constraint of water demand, the model response was not to crop in rainfed areas. These areas were cropped, but the yield reduction stood between 12% and 61%. The linear programming response at the municipal farm level reinforces the possibility of a continuous cropping system in irrigated areas and a double or single cropping system over rainfed areas. Analyzing the model responses in more depth, we should declare the limitations and potentials of the proposed integration for the study region.

The proposed LPM has limitations. First, being a uni-objective model, with this formulation, it is not possible to optimize other essential points for agricultural water management, such as increasing crop yields or efficient water allocation. Nonetheless, this limitation opens the possibility of developing and comparing other optimization methodologies, such as multi-objective linear programming, making integrating the systems even more helpful. Second, agricultural systems have a linear response to some variables, such as crop yield response to water stress and variations in general cost. The nonlinearity of the variables implies that we cannot guarantee that the forecasts of net benefit from the activity will be the same under actual conditions. We must also declare that the biweekly period considered in all case studies impacts the model's assertiveness, given that, in reality, crop management decisions such as irrigation are taken daily. Third, the LPM application does not consider the spatial variation of climatic parameters, representing a limitation for governance in the highest and most extensive water levels as in the entire mesoregion. Furthermore, the Microsoft Excel®Solver tool, used to solve the linear programming model, has a severe limitation of 200 decision variables and 100 restrictions for each evaluation. Therefore, it was necessary to divide the evaluation into shorter periods to make the model's size compatible with the tool.

As for the potential of this integration, we can cite the possibility of evaluating the area's productive potential over rainfall delay and reduction scenarios occasioned by climate change. The effects of rainfall delay significantly affect the double-cropping system practiced over rainfed areas, as demonstrated by Abrahão and Costa [15]. With the advanced information on the hydroclimatic system, we could apply the proposed model to simulate crops and variety with different water stress tolerance over the expected climate scenarios. Another consideration of the integration is to simulate the economic impacts of increasing or decreasing the water grants and, consequently, the irrigated area over a region. Analyzing the model results, especially at the municipal level, it becomes possible to estimate the labor, machinery, and capital needs, guiding job creation, industry strengthening, and funding actions for farmers. Finally, being simple and requiring few

input variables, the model can be easily understood and quickly disseminated to farmers, institutions, and decision makers that contribute to the water management process.

*Strategy for LPM Effectiveness*

The following actions are recommended to ensure that the model presented is effectively used as a water resources management tool in Western Bahia: (1) Develop and make available an online tool with the presented linear programming model, which can be made available on the OBahia platform [17] integrating water availability data and the rainy season onset forecast; (2) adjust the technical crop coefficients (need of resources, yield, and *Kc* for example) for regional conditions and make them available by default on the platform; (3) promote training for farmers and other decisions makers aiming at the dissemination and correct use of the online tool, demonstrating its potential for use and the possibilities of analysis.

## 6. Conclusions

Concerns about the water crisis and its environmental, social, and economic impacts on agricultural activity have increased in recent years, especially in regions where irrigated agriculture has extensive development, such as Western Bahia, Brazil. In this region, a hydroclimatic monitoring system was implemented, which, through data availability, aims to avoid water conflicts over water use, especially in periods of greater water scarcity. Considering this monitoring system, it became necessary to develop integrated tools that combine the water availability data and optimization tools, generating a valuable guide for water management actions. This study proposes a linear programming model that can be used for this purpose. The model maximizes the net benefit of agricultural activity considering land, labor, capital, machinery, and water restrictions. Its application was demonstrated at farm and municipality levels of water use and with different simulated rainfall delay and reduction conditions. The results show that the model response can guide decision making, especially when the adoption of double or continuous cropping systems is expected. For farmers, the model can provide information about which crop would be the best option for sowing and when to do so, considering the rainfall delay or reduction scenario given by the hydroclimatic forecast. The application can also estimate the economic impacts of new water grants, which is helpful for regional water authorities. Furthermore, the presented integration between a linear programming model and a hydroclimatic monitoring system is a scientific novelty. Integrating a linear programming model and a hydroclimatic monitoring system can better guide water management decisions.

**Supplementary Materials:** The following supporting information can be downloaded at: https://www.mdpi.com/article/10.3390/w14223625/s1, Spreadsheets S1: adopted coefficients.

**Author Contributions:** Conceptualization, project administration and funding acquisition, M.H.C.; field data collection, I.B., E.C.M. and A.G.d.S.J.; formal analysis, investigation and data curation, I.B.; writing—original draft preparation, I.B.; writing—review and editing, I.B., E.C.M., M.H.C. and A.G.d.S.J. All authors have read and agreed to the published version of the manuscript.

**Funding:** This research was funded by Coordination for the Improvement of Higher Education Personnel (CAPES), Finance Code: 001, by PRODEAGRO, grant numbers 011/2016 and 045/2019, and by the National Council for Scientific and Technological Development—Brazil (CNPq), process 132424/2019-3 and 441.210/2017-1.

**Data Availability Statement:** Example of LPM application and instructions are available on the OBahia Platform (http://obahia.dea.ufv.br/#/ruralprofit, accessed on 7 July 2020).

**Acknowledgments:** Gustavo B. Braga has read and commented on previous versions of this manuscript. We are thankful for his time and expertise. We also thank the farmers from Western Bahia and AIBA for providing data and support during data acquirement.

**Conflicts of Interest:** The authors declare no conflict of interest. The funders had no role in the design of the study.

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
