# Peer review of "A Linear Programming Model for Operational Optimization of Agricultural Activity Considering a Hydroclimatic Forecast—Case Studies for Western Bahia, Brazil"

_water, doi:10.3390/w14223625_

Round 1

Reviewer 1 Report

Summary: The paper provides an application of a mathematical programming model in order to simulate the behavior of farmers in conditions of uncertainty of the water resource. The analysis is carried out over the Western Bahia irrigated area in Brazil. It shows the potential of the tool in guiding farmers decision making process aimed at maximizing the net benefit of each agricultural activity in condition of water regime alteration.

Contribution: The topic addressed in the paper is relevant to the objectives and purposes of the journal, dealing with issues of water resource management and water scarcity (aims and scope are fitted). However, the paper is rather site-specific, and it is not clear how can contribute to advance in the field in a broader extent.   More detailed comments:   ·         Abstract: The overall structure is coherent with the scientific criteria of abstract definition.
However, it shows a low level of consistency due to the unclear statement of the research question. It should be better specified the research objective and emphasize the contribution relevance.
·         Introduction: it is unclear how this work should fit on the frontier of scientific literature (which is the element of novelty?). In addition, the analysis of the literature should be deepened and contextualized with respect to the international scenario, not only in Brazil (the references referred only to Brazilian context). In the light of a more in-depth scientific background, clearly formulate the research question and the hypothesis to be tested.

It seems that the advantage lies in the method used (linear programming) (although the authors point out that literature is full of works that have used this tool to optimize the economic benefit and allocate resources efficiently). I would suggest to describe the importance of guiding farmers in conditions of uncertainty by emphasizing the negative effects of the uncertainty of water availability in agriculture (in general, not only in Brazil) and therefore the usefulness of having information of this type. stress the strength of the work. the availability of climatic data used to implement the model (far from negligible as in this type of applications) It could be stressed the strenght of the work: the availaibility of weather data used to implement the model (far from negligible in this type of applications)   From line 34 to 38. The specifications about the characteristics of the case study must be reported directly in the methodology phase, as specified by the author himself.   ·         Materials and methods: In the description of the region give some more details on the two areas analyzed. How are the two areas characterized? (Type of crops grown, percentage of irrigated area, type of irrigation service, how the water resource is managed by the authority) Line 116. Bee more precise on the type of "weather data" used, in this way the reader can also have an idea of ​​the nature of the variables. Author affirms that variables fit with a normal distribution but to reference to statistics tests are reported. Line 126. Before explaining the components of a mathematical programming model, it is necessary to explain the theoretical basis on which the model is based. what are the theoretical assumptions of a LP model, how does it work? what is it made of? Then the characteristics of the components could be explained. Line 109. Some briefly explanation about the meaning of water grant are needed in order to understand how is managed the water resource in Western Bahia. Line 200. “Data are representative of the Bahia region”. are there any supporting statistics?

·         Discussion and Conclusions: Well-structured the discussion with general analysis followed by potential and limits of the model. However, both in discussion and in conclusion there is not directly addressed how these results could improve the water resource management (both at farm level and at municipalities level). How could the results orient the regional water authorities in improving the water resources management?

Author Response

We thank Reviewer #1 for pointing out the following topics.

1) Summary: The paper provides an application of a mathematical programming model in order to simulate the behavior of farmers in conditions of uncertainty of the water resource. The analysis is carried out over the Western Bahia irrigated area in Brazil. It shows the potential of the tool in guiding farmers decision making process aimed at maximizing the net benefit of each agricultural activity in condition of water regime alteration.

Response: We agree that the general idea of the work was completely understood by the Review #1.

2) Contribution: The topic addressed in the paper is relevant to the objectives and purposes of the journal, dealing with issues of water resource management and water scarcity (aims and scope are fitted). However, the paper is rather site-specific, and it is not clear how can contribute to advance in the field in a broader extent.

Response: We recognize and agree that more detail should be provided on how our study can contribute to the advancement of the field broadly. It is a fact that the linear programming model (LPM) presented was built with the objective of optimizing agricultural activity based on the local Hydroclimatic Forecast available for the region. Despite that, by providing the model example and a usage guide (Page 19, Lines 423-424), we enable another researcher to use this method and adapt it to other conditions and locations.

3) Abstract: The overall structure is coherent with the scientific criteria of abstract definition. However, it shows a low level of consistency due to the unclear statement of the research question. It should be better specified the research objective and emphasize the contribution relevance.

Response: We emphasize the research question, objectives, and relevance in abstract section (Page 1, Lines 13-17).

4) Introduction: it is unclear how this work should fit on the frontier of scientific literature (which is the element of novelty?). In addition, the analysis of the literature should be deepened and contextualized with respect to the international scenario, not only in Brazil (the references referred only to Brazilian context). In the light of a more in-depth scientific background, clearly formulate the research question and the hypothesis to be tested.

Response: We acknowledge and agree that more details should be provided on how our study fits in the frontier of scientific literature. We have included some international background about the water crisis situation (Page 1 Lines 35-39). We also combined the second and third paragraph (Pages 1-2, Lines 40-58). Finally, we clarify the research question (Page 2, Lines 82-85).

5) It seems that the advantage lies in the method used (linear programming) (although the authors point out that literature is full of works that have used this tool to optimize the economic benefit and allocate resources efficiently). I would suggest to describe the importance of guiding farmers in conditions of uncertainty by emphasizing the negative effects of the uncertainty of water availability in agriculture (in general, not only in Brazil) and therefore the usefulness of having information of this type. stress the strength of the work. the availability of climatic data used to implement the model (far from negligible as in this type of applications) It could be stressed the strenght of the work: the availaibility of weather data used to implement the model (far from negligible in this type of applications) From line 34 to 38. The specifications about the characteristics of the case study must be reported directly in the methodology phase, as specified by the author himself.

Response: We agree that the changes made in the previous question also apply to this.

6) Materials and methods: In the description of the region give some more details on the two areas analyzed. How are the two areas characterized? (Type of crops grown, percentage of irrigated area, type of irrigation service, how the water resource is managed by the authority) Line 116. Bee more precise on the type of "weather data" used, in this way the reader can also have an idea of ​​the nature of the variables. Author affirms that variables fit with a normal distribution but to reference to statistics tests are reported. Line 126. Before explaining the components of a mathematical programming model, it is necessary to explain the theoretical basis on which the model is based. what are the theoretical assumptions of a LP model, how does it work? what is it made of? Then the characteristics of the components could be explained. Line 109. Some briefly explanation about the meaning of water grant are needed in order to understand how is managed the water resource in Western Bahia. Line 200. “Data are representative of the Bahia region”. are there any supporting statistics?

Response: We acknowledge and agree that more details should be provided about the vale and Cerrado region. We have included more information on region description section (Page 4, Lines 108-116). We corrected the term “climatic” to “weather” when appropriated in all manuscript (Page 5, Lines 127, 129, 141, and 143), added a description of variables (Page 5, Lines 134,136). It was an error to affirm that variables fit a normal distribution, in fact the rainfall data were analyzed and for each fortnight period were obtained the average and quantiles values. We corrected the paragraph (Page 5, Lines 137-150). We have added the theoretical basis of adopted LPM and a brief explanation of the basis (Page 5, Lines 166-169). A brief explanation about water grant in Western Bahia was provided (Page 4, Lines 113-116). There are no statistics to confirm that the Duration, Initial Kc, Average Kc, and Final Kc are representatives. These data were obtained by local interviewers to Valley Scheduling advisors. In fact, this is not a limitation of our work, considering that in a practical application of this LPM the input variables should be determined by each farm or region, depending on the local conditions. The sentence was rewritten to avoid any misunderstood (Page 9, Lines 242-243).

7) Discussion and Conclusions: Well-structured the discussion with general analysis followed by potential and limits of the model. However, both in discussion and in conclusion there is not directly addressed how these results could improve the water resource management (both at farm level and at municipalities level). How could the results orient the regional water authorities in improving the water resources management?

Response: We agree that the presented linear programming model has a potential to improve the water resources management in Western Bahia, been necessary a better description about. To clear this topic on discussion section, we kept the consideration in first part of the section (Page 17, Lines 347-361), included more information at LPM potentials subsection (Page 17, Lines 371-375), and included a summary of benefits for farmers and regional water resources authorities at conclusions section (Page 17, Lines 409-413).

Reviewer 2 Report

The subject of the article is interesting, and it is linked to the objectives of the journal, however, there are some issues that have to be reconsidered.

Even if the Introduction part describes the domain, the literature review (results of similar studies etc.) is quite superficially discussed, and it must be reconsidered.

The structure of the article must be clearly presented in the Introduction part.

Also, the main objective and the secondary objectives/hypothesis are missing.

A better deeper explanation of why this sample is representative for the entire studied population could be helpful.

The results are interesting and they are well discussed, but the conclusions are not enough to sustain the results. The use of the research is, so, insufficiently explained at the conclusion part. It is advisable to create a distinct part for formulating general conclusions and recommendations for scholars, government, business etc.

Author Response

We thank Reviewer #2 for pointing out the following topics.

1) The subject of the article is interesting, and it is linked to the objectives of the journal, however, there are some issues that have to be reconsidered.

Response: We thank the Reviewer #2 for this comment.

2) Even if the Introduction part describes the domain, the literature review (results of similar studies etc.) is quite superficially discussed, and it must be reconsidered.

Response: We reconsidered the introduction section, adding a global statement of water crisis (Page 1, Lines 35-39), and declaring the problem statement (Page 2, Lines 82-85).

3) The structure of the article must be clearly presented in the Introduction part.

Response: We have included the structure of article at the end of introduction section (Page 2, Lines 90-93).

4) Also, the main objective and the secondary objectives/hypothesis are missing.

Response: We acknowledge and agree that the objectives of the work should be clearly presented. We have added the problem statement and reformulate the objectives of the work (Page 2, Lines 82-93).

5) A better deeper explanation of why this sample is representative for the entire studied population could be helpful.

Response: As the Reviwer #1 made a similar appointment we have reconsidered the region description section  (Page 3, Lines 108-116) and the crop characterization subsection (Page 9, Lines 242-245). The answer for Reviwer#1 was: We acknowledge and agree that more details should be provided about the vale and Cerrado region. We have included more information on region description section (Page 4, Lines 108-116). We corrected the term “climatic” to “weather” when appropriated in all manuscript (Page 5, Lines 127, 129, 141, and 143), added a description of variables (Page 5, Lines 134,136). It was an error to affirm that variables fit a normal distribution, in fact the rainfall data were analyzed and for each fortnight period were obtained the average and quantiles values. We corrected the paragraph (Page 5, Lines 137-150). We have added the theoretical basis of adopted LPM and a brief explanation of the basis (Page 5, Lines 166-169). A brief explanation about water grant in Western Bahia was provided (Page 4, Lines 113-116). There are no statistics to confirm that the Duration, Initial Kc, Average Kc, and Final Kc are representatives. These data were obtained by local interviewers to Valley Scheduling advisors. In fact, this is not a limitation of our work, considering that in a practical application of this LPM the input variables should be determined by each farm or region, depending on the local conditions. The sentence was rewritten to avoid any misunderstood (Page 9, Lines 242-243).

6) The results are interesting, and they are well discussed, but the conclusions are not enough to sustain the results. The use of the research is, so, insufficiently explained at the conclusion part. It is advisable to create a distinct part for formulating general conclusions and recommendations for scholars, government, business etc.

Response: We have reconsidered the discussion (Page 19, Lines 371-375) and conclusions section (Page 19, Lines 409-413).

Reviewer 3 Report

SUMMARY: The paper applies a linear programming (LP) approach to optimize agricultural production in Western Bahia, Brazil, taking into account projections of the start date of the rainy season from a prototype hydroclimatic forecasting system. The model is demonstrated with case studies of five farms and three municipalities.

OVERALL RECOMMENDATION: My overall recommendation is to reconsider the manuscript after major revisions. The manuscript has serious flaws that would need to be addressed for it to be valuable to the readers of water (see below). While there seems to be a possibly valuable modeling effort underlying the paper, the presentation is not sufficiently clear. Thematically, the research presented could be within the scope of Water.

RESEARCH QUESTION AND NOVELTY: No research question is explicitly stated in the introduction (Section 1), nor can one be easily inferred from the manuscript. This makes it difficult to follow the remainder of the presentation. The literature review does not clearly identify a research gap. Whether this paper provides a novel research contribution is, therefore, difficult to assess. This is particularly important since the first LP models of (irrigated) agricultural production have been developed a long time ago, and there is, by now, a wide literature. The innovation might stem from the combination with the particular type of hydroclimatic forecast used. The manuscript would need to demonstrate in which regard the model used or the insights gained go substantially beyond what already exists in the literature.

METHOD SELECTION: The fact that neither the research question nor the research gap are evident also makes it difficult to evaluate whether the method chosen is appropriate. Well-known limitations of LP models, such as overspecialization should be discussed in terms of their relevance for the analysis presented, and with reference to existing discussions in the literature, in order to justify the selected method, compared to other options. Section 5.2 mentions LP limitations, but only addresses some very general points (e.g. LP is not able to reflect non-linear crop yield responses to water stress) and does not elaborate on what that could mean for the findings of the present analysis.

METHOD DOCUMENTATION: The description of the method itself in Sections 2 and 3  seems mostly appropriate, though some more detail on the constraint data used would helpful to ensure replicability. The description of the scenarios and model assumptions should be related to the research question. The model description should make explicit whether there are any differences between the farm and the municipal model besides different parameter inputs, or not.

RESULT PRESENTATION: The lack of a clearly defined research question particularly hampers the presentation and discussion of results in Sections 4 and 5. The presentation of results in Section 4 is detailed and suggests potentially interesting insights. However, it is not sufficiently clear which results are specific to the LP model, and which could be directly derived from the model inputs. For example, all irrigated areas except for Sama and Correntina seem to be fully planted across scenarios. Is this simply because there is sufficient irrigation water to balance the rainfall reductions and delays analyzed? Would the LP model be needed to understand this? Similarly, it is unclear whether the delayed cropping in the rainfed areas is simply a direct reflection of the model inputs making it impossible to plant earlier, or whether more interesting mechanisms are at work. And why are, e.g., all 2020 rainfed crops in Barreiras shifted back by the same number of fortnights in scenarios D2-D4? Rather than going through the detail of one case after the other, the manuscript would strongly benefit from presenting the results and the underlying mechanisms systematically, based on a clear research question.

RESULT DISCUSSION: The result discussion in Section 5 would likewise benefit from being clearly related to a research question. The Discussion section currently evaluates the model as successful because it maximized the economic benefit of agricultural activity across different levels of water use and different administrative scales. It remains unclear how the analysis presented shows that. Any LP model that has economic benefit as an objective function would by assumption maximize it within the given constraints, as long as the inputs allow for any feasible solution. So here the manuscript seems to assume what it presents as a conclusion. The remainder of the discussion seems to go beyond the presented results. For example, evaluating the areas productive potential and guiding water resource governance are mentioned. Both sound interesting, but are not discussed explicitly: What can we conclude for the productive potential of the area under climate change? Is it surprisingly high or low, and by what measure? What implications for water resource governance would follow from the presented results? These are just examples of what a substantive discussion of presented results should entail.

Sections 5.1.-5.3. should then build on this. After hardly discussing the results at all in the first paragraph of Section 5, Section 5.1. immediately discusses future applications of this LP model or its extensions. If the results presented are interesting enough, their implications should be discussed. Otherwise, it might be worth considering to add some of the analyses listed here to the results presented in the paper. Section 5.2. describes general LP limitations, such as having linear response functions, without addressing how this should be factored in when interpreting the results presented here.

ABSTRACT AND CONCLUSION: The Abstract and Conclusion reflect the overall shortcomings of the current manuscript. “Results show that the linear programming model can provide information to optimize the agricultural activity by recommending crops, seeding dates, and cycle intervals, predicting harvest losses, and optimizing resources allocations.” A revised version of the manuscript would need to be explicit about what these recommendations, losses, and optimized allocations are (quantitatively, or at least qualitatively), and in which way the LP model has allowed for novel insights that were not possible before the innovative analysis presented in the paper.

INDIVIDUAL LINE COMMENTS: I recommend additional proof-reading of the manuscript to correct grammatical errors and improve unclear formulations, some examples of which are listed below.

ABSTRACT: Language of the abstract features numerous unclear formulations and grammatical errors, making it difficult for the reader to understand what the paper is about:

- Lines 11-13: Too vague: The sentences express that information is needed to rationally use water resources and that decision-making is necessary to avoid conflicts between water uses. Without specifying the type of information and decisions addressed, it remains unclear which information these sentences are meant to convey.

- Line 14: “By concern about the water crisis in region, an hydroclimatic monitoring system […]” -> “Due to concerns about the water crisis in the region, a hydroclimatic monitoring system”

- Lines 14-15: The description of the motivation also remains too vague: “a hydroclimatic monitoring system was proposed […] Considering that, it becomes necessary to develop a tool capable to use this information providing a guide to farmers and regional water managers to better decisions.” Was the monitoring system just proposed or implemented? Why does this proposal make it “necessary to develop” the model presented here?

- Lines 19-20: What is new about the results? “Results show that the linear programming model can provide information to optimize the agricultural activity […]” – is that really a new insight gained from the results? Isn’t that true for agricultural LP models in general, by their structure? How do the results demonstrate this?

1. INTRODUCTION:

- Line 35: “have been increase.” -> “have been increasing”

- Lines 61-64: It remains unclear what this study aims to improve: “Impact strongly” in which regard? To realize which “potential”? Why is this “necessary”? Is this just about enhancing profitability or additional objectives? If so, how are these operationalized?

- Lines 75-76: “In summary, LP is commonly used in agricultural management, with a direct relationship between model robustness and data reliability and availability.” - This statement seems quite generic. After reviewing the literature, it would have been helpful to identify a research gap that this study aims to address.

2. MATERIALS AND METHODS:

- Figure 1: The figure is located in a part of the paper where it is shortly mentioned, but hardly discussed. I recommend moving this to section 2.2., where dataset acquisition is described.

- Figure 1: Double-check the flow-chart elements, esp. if the rhombus elements are used correctly and if “INMET/CONAB/…” should be in boxes.

2.1. REGION DESCRIPTION:

- Line 100: Is AIBA a neutral source in judging which areas should be used for small-scale versus commercial agriculture? If this point is relevant to the research and presented here, please make sure to provide a balanced view of the competing potential land and water uses.

- Line 112: Define “Q90” for the benefit of the reader

2.3. LPM FORMULATION:

- Eq. 4 : CWD does not seem to feed back into the objective function. Is water cost free?

6. CONCLUSIONS:

- Line 388: “appliable" -> “applicable”

Author Response

Reviewer #3

We thank Reviewer #3 for the excellent comments on the manuscript. We hope to have answered all the notes made.

SUMMARY: The paper applies a linear programming (LP) approach to optimize agricultural production in Western Bahia, Brazil, taking into account projections of the start date of the rainy season from a prototype hydroclimatic forecasting system. The model is demonstrated with case studies of five farms and three municipalities.

OVERALL RECOMMENDATION: My overall recommendation is to reconsider the manuscript after major revisions. The manuscript has serious flaws that would need to be addressed for it to be valuable to the readers of water (see below). While there seems to be a possibly valuable modeling effort underlying the paper, the presentation is not sufficiently clear. Thematically, the research presented could be within the scope of Water.

Authors’ Response: We appreciate the notes and hope that the new version of the manuscript meets all these considerations.

RESEARCH QUESTION AND NOVELTY: No research question is explicitly stated in the introduction (Section 1), nor can one be easily inferred from the manuscript. This makes it difficult to follow the remainder of the presentation. The literature review does not clearly identify a research gap. Whether this paper provides a novel research contribution is, therefore, difficult to assess. This is particularly important since the first LP models of (irrigated) agricultural production have been developed a long time ago, and there is, by now, a wide literature. The innovation might stem from the combination with the particular type of hydroclimatic forecast used. The manuscript would need to demonstrate in which regard the model used or the insights gained go substantially beyond what already exists in the literature.

Authors’ Response: We thank the Reviewer #1 for this appointment. The topic was already considered in previous revisions, so we reconsidered the whole abstract, introduction, and conclusion sections. The research gap is the integration between the hydroclimatic monitoring system and the linear programming method.

METHOD SELECTION: The fact that neither the research question nor the research gap are evident also makes it difficult to evaluate whether the method chosen is appropriate. Well-known limitations of LP models, such as overspecialization should be discussed in terms of their relevance for the analysis presented, and with reference to existing discussions in the literature, in order to justify the selected method, compared to other options. Section 5.2 mentions LP limitations, but only addresses some very general points (e.g. LP is not able to reflect non-linear crop yield responses to water stress) and does not elaborate on what that could mean for the findings of the present analysis.

Authors’ Response: In the revised version of the manuscript, the research gap and question were declared in a better way. The research gap and question refer to the integration between the linear programming method and the hydroclimatic forecast system. The limitations of the model are further discussed in subsection 5.2.

METHOD DOCUMENTATION: The description of the method itself in Sections 2 and 3  seems mostly appropriate, though some more detail on the constraint data used would helpful to ensure replicability. The description of the scenarios and model assumptions should be related to the research question. The model description should make explicit whether there are any differences between the farm and the municipal model besides different parameter inputs, or not.

Authors’ Response: The scenarios were defined as possible responses to the hydroclimatic forecast into rainfall delay or reduction. These scenarios demonstrate the model’s potential, considering that the input data will come from the hydroclimatic forecast in fundamental analysis. We highlight this point at Pag 25 Lines 171-172. All differences between the model application are described in the Case Studies section.

RESULT PRESENTATION: The lack of a clearly defined research question particularly hampers the presentation and discussion of results in Sections 4 and 5. The presentation of results in Section 4 is detailed and suggests potentially interesting insights. However, it is not sufficiently clear which results are specific to the LP model, and which could be directly derived from the model inputs. For example, all irrigated areas except for Sama and Correntina seem to be fully planted across scenarios. Is this simply because there is sufficient irrigation water to balance the rainfall reductions and delays analyzed? Would the LP model be needed to understand this? Similarly, it is unclear whether the delayed cropping in the rainfed areas is simply a direct reflection of the model inputs making it impossible to plant earlier, or whether more interesting mechanisms are at work. And why are, e.g., all 2020 rainfed crops in Barreiras shifted back by the same number of fortnights in scenarios D2-D4? Rather than going through the detail of one case after the other, the manuscript would strongly benefit from presenting the results and the underlying mechanisms systematically, based on a clear research question.

 “The lack of a clearly defined research question particularly hampers the presentation and discussion of results in Sections 4 and 5.”

Authors’ Response: As mentioned in previous comments, the research question was reformulated in the abstract and introduction sections to reinforce this point. 

 “ The presentation of results in Section 4 is detailed and suggests potentially interesting insights. However, it is not sufficiently clear which results are specific to the LP model, and which could be directly derived from the model inputs.”

 Authors’ Response: The model formulation and the input data impact the results. Input data, such as the evaluated scenarios, generate the conditions analyzed in the modeling. Likewise, the presented model optimizes a response based on the given conditions. This further reinforces the importance of this integrated model in hydroclimatic forecasting. The results of the water availability forecast in the study region are fundamental data for planning the use of water resources. However, without an analysis tool like the model presented, we do not have an efficient way to transform this data into useful information.

 “For example, all irrigated areas except for Sama and Correntina seem to be fully planted across scenarios. Is this simply because there is sufficient irrigation water to balance the rainfall reductions and delays analyzed? Would the LP model be needed to understand this?”

Authors’ Response: In a superficial analysis, we do not need a linear programming model to demonstrate that we can cultivate the most profitable crop for the activity's success in areas where water is available for irrigation. The problem is that with the recurring scenarios of reduced water availability, we cannot guarantee the availability of water for all crops. The water grant follows the federal legislation mentioned in the introduction, which gives priority use to human consumption, the watering of animals, and other uses, such as agriculture. In addition to being made for a limited time, the renewal depends on analysis. It is a preoccupation considering the region's recurrent scenarios of rainfall instability. Another point is that the model also considers other constraints besides water availability for optimization, such as the availability of labor and machinery on the farm.

“Similarly, it is unclear whether the delayed cropping in the rainfed areas is simply a direct reflection of the model inputs making it impossible to plant earlier, or whether more interesting mechanisms are at work. And why are, e.g., all 2020 rainfed crops in Barreiras shifted back by the same number of fortnights in scenarios D2-D4?”

Authors’ Response: Again, we state that the input data and the model formulation impact the response. For example, if we had only one option for the rainfed crop with an expected yield. In this case, the recommendation to start sowing would accompany the beginning of the rains in the region. However, the formulation of the model presented seeks to optimize the net benefit of the activity, considering the various crop options and restrictions. Thus, as in Barreiras, instead of considering the beginning of the rain forecast based on restrictions, the model recommended the cultivation of another crop, which would be more profitable.

“Rather than going through the detail of one case after the other, the manuscript would strongly benefit from presenting the results and the underlying mechanisms systematically, based on a clear research question.”

Authors’ Response: We believe that the way the manuscript was presented, we have gained in presenting the results of each case study and subsequently having a discussion based on the research question. That is why we chose to maintain this structure for presenting the results.

RESULT DISCUSSION: The result discussion in Section 5 would likewise benefit from being clearly related to a research question. The Discussion section currently evaluates the model as successful because it maximized the economic benefit of agricultural activity across different levels of water use and different administrative scales. It remains unclear how the analysis presented shows that. Any LP model that has economic benefit as an objective function would by assumption maximize it within the given constraints, as long as the inputs allow for any feasible solution. So here the manuscript seems to assume what it presents as a conclusion. The remainder of the discussion seems to go beyond the presented results. For example, evaluating the areas productive potential and guiding water resource governance are mentioned. Both sound interesting, but are not discussed explicitly: What can we conclude for the productive potential of the area under climate change? Is it surprisingly high or low, and by what measure? What implications for water resource governance would follow from the presented results? These are just examples of what a substantive discussion of presented results should entail.

Authors’ Response: We reconsider the discussion section regarding the research question.

Sections 5.1.-5.3. should then build on this. After hardly discussing the results at all in the first paragraph of Section 5, Section 5.1. immediately discusses future applications of this LP model or its extensions. If the results presented are interesting enough, their implications should be discussed. Otherwise, it might be worth considering to add some of the analyses listed here to the results presented in the paper. Section 5.2. describes general LP limitations, such as having linear response functions, without addressing how this should be factored in when interpreting the results presented here.

Authors’ Response: We reconsider the discussion section considering this appointment.

ABSTRACT AND CONCLUSION: The Abstract and Conclusion reflect the overall shortcomings of the current manuscript. “Results show that the linear programming model can provide information to optimize the agricultural activity by recommending crops, seeding dates, and cycle intervals, predicting harvest losses, and optimizing resources allocations.” A revised version of the manuscript would need to be explicit about what these recommendations, losses, and optimized allocations are (quantitatively, or at least qualitatively), and in which way the LP model has allowed for novel insights that were not possible before the innovative analysis presented in the paper.

Authors’ Response: We reconsider the abstract and conclusion section based on this appointment. 

INDIVIDUAL LINE COMMENTS: I recommend additional proof-reading of the manuscript to correct grammatical errors and improve unclear formulations, some examples of which are listed below.

Authors’ response: The entire manuscript passed by an extensive grammatical correction.

ABSTRACT: Language of the abstract features numerous unclear formulations and grammatical errors, making it difficult for the reader to understand what the paper is about:

- Lines 11-13: Too vague: The sentences express that information is needed to rationally use water resources and that decision-making is necessary to avoid conflicts between water uses. Without specifying the type of information and decisions addressed, it remains unclear which information these sentences are meant to convey.

Authors’ response: We thank the reviewer for this appointment. Considering that, we reformulated the abstract section.

- Line 14: “By concern about the water crisis in region, an hydroclimatic monitoring system […]” -> “Due to concerns about the water crisis in the region, a hydroclimatic monitoring system”

Authors’ response: Modification is done.

- Lines 14-15: The description of the motivation also remains too vague: “a hydroclimatic monitoring system was proposed […] Considering that, it becomes necessary to develop a tool capable to use this information providing a guide to farmers and regional water managers to better decisions.” Was the monitoring system just proposed or implemented? Why does this proposal make it “necessary to develop” the model presented here?

Authors’ response: The hydroclimatic monitoring system was implemented, and the term was corrected. Also, the need for the linear programming model developed was presented.

- Lines 19-20: What is new about the results? “Results show that the linear programming model can provide information to optimize the agricultural activity […]” – is that really a new insight gained from the results? Isn’t that true for agricultural LP models in general, by their structure? How do the results demonstrate this?

Authors’ response: It was unclear that the novelty is the integration between the linear programming model and the hydroclimatic monitoring system. We reconsidered the abstract structure to make it notable.

  1. INTRODUCTION:

- Line 35: “have been increase.” -> “have been increasing”

Authors’ response: Considering that it was recommended that “Moderate English changes” in our manuscript, we reviewed this paragraph and corrected the structure.

- Lines 61-64: It remains unclear what this study aims to improve: “Impact strongly” in which regard? To realize which “potential”? Why is this “necessary”? Is this just about enhancing profitability or additional objectives? If so, how are these operationalized?

Authors’ response: We reconsider the paragraph structure to present the integration between the hydroclimatic monitoring system and the tool as a need for this region, avoiding the available data becoming obsolete. The following paragraph presents this integration between systems as the research gap.

- Lines 75-76: “In summary, LP is commonly used in agricultural management, with a direct relationship between model robustness and data reliability and availability.” - This statement seems quite generic. After reviewing the literature, it would have been helpful to identify a research gap that this study aims to address.

Authors’ response: The paragraph's final part was modified to present the research gap.

  1. MATERIALS AND METHODS:

- Figure 1: The figure is located in a part of the paper where it is shortly mentioned, but hardly discussed. I recommend moving this to section 2.2., where dataset acquisition is described.

Authors’ response: We thank Reviewer #3 for this significant consideration. Subsection 2.2 explicitly describes the process of acquiring and processing the data. However, this figure's objective is to visually expose the work to the reader, in a visual way, the entire methodological process adopted. The figure also shows the model formulation, case studies, analysis and discussion of the results, and topics not described in subsection 2.2. We chose to keep this figure in its original form as we believe that viewing the entire process before reading it can improve the understanding of the methods adopted.

- Figure 1: Double-check the flow-chart elements, esp. if the rhombus elements are used correctly and if “INMET/CONAB/…” should be in boxes.

Authors’ response: Figure 1 was reviewed, and the elements were corrected.

2.1. REGION DESCRIPTION:

- Line 100: Is AIBA a neutral source in judging which areas should be used for small-scale versus commercial agriculture? If this point is relevant to the research and presented here, please make sure to provide a balanced view of the competing potential land and water uses.

Authors’ response: AIBA is the association of farmers and irrigators in Western Bahia. This association is not responsible for delimiting the areas of small or large-scale agriculture. Declaring that exists two predominant agricultural models, and this is in different regions was the objective of this phrase. Since this information is not highly relevant to the research, we removed this phrase to avoid misunderstandings.

- Line 112: Define “Q90” for the benefit of the reader

Authors’ response: Q90 was described as a “minimum amount of water flow in a river that is present 90% of the time”.

2.3. LPM FORMULATION:

- Eq. 4 : CWD does not seem to feed back into the objective function. Is water cost free?

Authors’ response: Yes, there is no water cost for irrigation in Brazil.

  1. CONCLUSIONS:

- Line 388: “appliable" -> “applicable”

Authors’ response: Modification is done.

Reviewer 4 Report

The authors propose a PL model to provide information on land use and management of production activities.

A mathematical programming model simplifies reality, but in the present research the model is too simplified: one objective function and three constraints to define and program something that is very complex.

Author Response

1) The authors propose a PL model to provide information on land use and management of production activities. A mathematical programming model simplifies reality, but in the present research the model is too simplified: one objective function and three constraints to define and program something that is very complex.

Author's Response: We thank Reviewer #4 for this comment. When we compare this model to the others mentioned in the manuscript's introduction section, we realize there is a simplification of the process. Despite being simplified, the model is robust and generates reliable answers for managing water resources in the region. We can also say that the great novelty of the work is related to integrating the optimization model with the hydroclimatic forecast of the region. This point was highlighted in the revised version of the manuscript. This integration is what allows a simplification of the optimization model. We do not rule out the development, testing, and comparison of other optimization models, but this step will be reported in future works.

Reviewer 5 Report

Question 1: The author should pay attention to the format of the article, such as the expression of the year and the unification of the references.

Question 2: In the abstract, the author proposes an hydroclimatic monitoring system. There is no specific explanation in the article.

Question 3: The author used linear programming method to optimize the agricultural output in irrigated and rainfed agricultural areas, but compared the change of planting area under different rainfall reduction or delay conditions in different regions with LPM model.

Question 4: According to the author's research results, the LPM model results have little impact on crops under the condition of reduced rainfall and the original agricultural activities. Is it similar in rainfed areas?

Question 5: The division of rainfall reduction and rainfall delay gradient lacks practical basis. The crop growth period selected in this paper is generally 4-6 months. The deviation of rainfall reduction and rainfall delay set by the author is too large.

Question 6: The illustration should be clear, and the illustration of figure 8 in the article is too vague.

Question 7: The linear model is used to optimize non-linear agricultural activities. The method adopted by the author has great limitations. Please carefully refer to the practical conditions of previous research methods.

Author Response

Reviewer #5

We thank Reviewer #5 for pointing out the following topics.

Question 1: The author should pay attention to the format of the article, such as the expression of the year and the unification of the references.

Authors' Response: The manuscript passed by a grammatical and formatting revision, and this error was corrected.

Question 2: In the abstract, the author proposes an hydroclimatic monitoring system. There is no specific explanation in the article.

Authors' Response: In the abstract section, It was cited that the region's hydroclimatic monitoring system is already implemented. The novelty proposed in this article is the integration between the proposed linear programming method and the hydroclimatic forecast that already exists. The hydroclimatic monitoring system references counts in the fourth paragraph of the introduction section.

Question 3: The author used linear programming method to optimize the agricultural output in irrigated and rainfed agricultural areas, but compared the change of planting area under different rainfall reduction or delay conditions in different regions with LPM model.

Authors' Response: In problem formulation, the objective function maximizes the net economic benefit considering the decision variables as the planted area of a crop, over the irrigated or rainfed condition, with a specific crop cycle, produced in a particular fortnight. The possibility of annual variation of rainfall conditions predicted with the hydroclimatic made to simulate the LPM response over the condition of rainfall delay and reduction. Due to the problem formulation, the variation between cultivated areas and crops is expected.

Question 4: According to the author's research results, the LPM model results have little impact on crops under the condition of reduced rainfall and the original agricultural activities. Is it similar in rainfed areas?

Authors’ response: Considering the formulation of the problem in this work, if the rainfall scenarios meet the water demand of the crop that generates the highest net benefit from the activity, we will not have a difference in the response of the model. We want to establish that the water demand adopted here considers a reduction in ETc that does not significantly impact crop productivity (Kr), evaluated for the region of this study by the authors cited in reference 44.

Question 5: The division of rainfall reduction and rainfall delay gradient lacks practical basis. The crop growth period selected in this paper is generally 4-6 months. The deviation of rainfall reduction and rainfall delay set by the author is too large.

Authors' Response: The adopted scenarios came from analysis to simulate possible rainfall scenarios of delay and reduction. The proposal to demonstrate the linear programming method's response does not affect the lower lacks' results.

Question 6: The illustration should be clear, and the illustration of figure 8 in the article is too vague.

Authors' response: The illustration in Figure 8 represents the same analysis made at the farm level but at the municipal level.

Question 7: The linear model is used to optimize non-linear agricultural activities. The method adopted by the author has great limitations. Please carefully refer to the practical conditions of previous research methods.

Authors Response: Adapting the objective function and constraints allows us to use a linear programming method in our study. For example, the crop yield response to water does not follow a linear response, but we adapt this constraint, using the Kr coefficient, a limit for ETc reduction, to make this response linear. This limitation is cited in 5.2 subsection.

Round 2

Reviewer 1 Report

I can see some improvement in revised version. However, the paper is rather site-specific, and it is not clear how can contribute to advance in the field in a broader extent. Anfortunately, Authors have not deepen this point.

Author Response

Reviewer #1 

We thank Reviewer #1 for pointing out the following topics.

1) I can see some improvement in revised version. However, the paper is rather site-specific, and it is not clear how can contribute to advance in the field in a broader extent. Anfortunately, Authors have not deepen this point.

Response: We acknowledge and agree that how our work can contribute to advance in a broader extent. So, we have reformulated the conclusion section (Page 18, Lines 397-416) in following themes: (1) Brief contextualization about the water crisis and its impacts; (2) the hydroclimatic forecast and needs of tool to make the information useful; (3) the solution presented in work; (4) the uses and implications for Western Bahia and other regions; (5) the possibility of application of this model in other regions and conditions by the scientific community.

Reviewer 2 Report

The authors improved the article in a way that is acceptable for being published,

Author Response

Reviewer #2 

We thank Reviewer #2 for pointing out the following topics.

1) The authors improved the article in a way that is acceptable for being published,

Response: We thank the Reviewer #2 for this and previous comments.

Reviewer 5 Report

I think this manuscript has met the requirements of the journal, but the language needs to be improved.

Author Response

Reviewer #5 

Comment 1: I think this manuscript has met the requirements of the journal, but the language needs to be improved.

Authors' Response: We thank the reviewer for the comment and for considering that we again did a grammatical and spelling review of the document.

Round 3

Reviewer 1 Report

As a whole, this research report a good application of LPM. While in Brasil this tool for crop optimisation could be seen novel, in the scientific arena has been developped 40 years ago. Also the possible application for planning and managing water resources exhibits long history. I think this kind of researches shoud be best placed within a national or local scientific arena. Otherwise, you need make your research innovative for all scientists, in the world.

I am sorry to confirm my previus judgment.

Author Response

Reviewer #1 

We thank Reviewer #1 for pointing out the following topics.

1) As a whole, this research report a good application of LPM. While in Brasil this tool for crop optimisation could be seen novel, in the scientific arena has been developped 40 years ago. Also the possible application for planning and managing water resources exhibits long history. I think this kind of researches shoud be best placed within a national or local scientific arena. Otherwise, you need make your research innovative for all scientists, in the world. I am sorry to confirm my previus judgment.

Response:

We thank the reviewer #1 for pointing this topic.

We agree that the LPM used in this work has been widely used by the scientific community, but we cannot disregard our progress given the reality of our work. The planning of water resources uses is a topic with a global relevance, given the impacts of climate change on the conditions of water availability. This point is also relevant in the study region of this work, the Western Bahia, where agriculture plays a great contribution in the economic and social fields.

In Western Bahia, water availability conditions throughout the year are uncertain, given the climate characteristics of short rainy season and the great variation in its onset. In this way, as mentioned in the work, a hydroclimatic monitoring system was developed, providing essential data of water availability. To make this data useful for decision-making process of water resources, we note the need to develop a tool for integrating these data with local information, allowing for assertive responses of which is the best agricultural decision for the region.

The great novelty of this work is precisely the integration of hydroclimatic monitoring data with local data from farmers and regional historical data to generate this assertive response. Other works, which were also reported in the introduction section, use robust models precisely when data are not available or when the uncertainties associated with them are considerable, which is not our reality. In addition, our work also enables a comparative analysis between irrigated and rainfed areas, in relation to crop yield, gross profit, land use throughout the year and use of water resources. In future work, we intend to apply other optimization methods and compare the results with this.

We agree that these topics have been highlighted in the manuscript. Considering the fulfillment of the previous comments, we ask the reviewer to reconsider this submission. The integration of the hydroclimatic forecasting system with local information is a scientific novelty and should be considered.

Round 4

Reviewer 1 Report

I am sure your research is relevant for the Western Bahia, thus such relevance should be better listed into a local or national Journal. Although, your model trys to cope with rain pattern uncertainity, from the analysis there is no evidence that such approach is better than conventional one. 

In any case, at this point I will not come over my recommendation. 

Author Response

Reviewer #1

We thank Reviewer #1 for the comment on the manuscript.

Comment 1: I am sure your research is relevant for the Western Bahia, thus such relevance should be better listed into a local or national Journal. Although, your model trys to cope with rain pattern uncertainity, from the analysis there is no evidence that such approach is better than conventional one. In any case, at this point I will not come over my recommendation. 

Authors' Response: We agree that the research is relevant to Western Bahia, but we will still consider submission to the Water MDPI, considering that the work falls within the scope of this journal. In addition, we also exemplify other papers about Western Bahia that were published in Water and other MDPI journals.

Pousa, R., Costa, M. H., Pimenta, F. M., Fontes, V. C., & Castro, M. (2019). Climate change and intense irrigation growth in Western Bahia, Brazil: The urgent need for hydroclimatic monitoring. Water, 11(5), 933. https://doi.org/10.3390/w11050933

Dionizio, E. A., & Costa, M. H. (2019). Influence of land use and land cover on hydraulic and physical soil properties at the cerrado agricultural frontier. Agriculture (Switzerland), 9(1), 24. https://doi.org/10.3390/agriculture9010024

Santos, A. B., Costa, M. H., Mantovani, E. C., Boninsenha, I., Castro, M., Heil Costa, M., Mantovani, C., Boninsenha, I., & Castro, M. (2020). A Remote Sensing Diagnosis of Water Use and Water Stress in a Region with Intense Irrigation Growth in Brazil. Remote Sensing, 12(22), 3725. https://doi.org/10.3390/rs12223725

Pimenta, F. M., Speroto, A. T., Costa, M. H., & Dionizio, E. A. (2021). Historical changes in land use and suitability for future agriculture expansion in Western Bahia, Brazil. Remote Sensing, 13(6), 1–31. https://doi.org/10.3390/rs13061088